# Sleep in Disorders of Consciousness: A Brief Overview on a Still under Investigated Issue

**DOI:** 10.3390/brainsci13020275

**Published:** 2023-02-07

**Authors:** Loredana Raciti, Gianfranco Raciti, David Militi, Paolo Tonin, Angelo Quartarone, Rocco Salvatore Calabrò

**Affiliations:** 1IRCCS Centro Neurolesi Bonino Pulejo, 98121 Messina, Italy; 2Sant’Anna Institute, 88900 Crotone, Italy

**Keywords:** disorders of consciousness, sleep, UWS and MCS, wake-sleep cycle

## Abstract

Consciousness is a multifaceted concept, involving both wakefulness, i.e., a condition of being alert that is regulated by the brainstem, and awareness, a subjective experience of any thoughts or perception or emotion. Recently, the European Academy of Neurology has published international guidelines for a better diagnosis of coma and other disorders of consciousness (DOC) through the investigation of sleep patterns, such as slow-wave and REM, and the study of the EEG using machine learning methods and artificial intelligence. The management of sleep disorders in DOC patients is an increasingly hot topic and deserves careful diagnosis, to allow for the most accurate prognosis and the best medical treatment possible. The aim of this review was to investigate the anatomo-physiological basis of the sleep/wake cycle, as well as the main sleep patterns and sleep disorders in patients with DOC. We found that the sleep characteristics in DOC patients are still controversial. DOC patients often present a theta/delta pattern, while epileptiform activity, as well as other sleep elements, have been reported as correlating with outcomes in patients with coma and DOC. The absence of spindles, as well as REM and K-complexes of NREM sleep, have been used as poor predictors for early awakening in DOC patients, especially in UWS patients. Therefore, sleep could be considered a marker of DOC recovery, and effective treatments for sleep disorders may either indirectly or directly favor recovery of consciousness.

## 1. Introduction

Consciousness is a multifaceted concept involving both wakefulness, i.e., a condition of being alert rather than sleepy that is regulated by the brainstem [1], and awareness, a subjective experience of any thoughts or perception or emotion [2]. The latter is divided into internal awareness, referring to any imagination, internalized/self-directed dialogue, or erratic mind; and external awareness, representing the perception of sensory stimuli [3,4].

Disorders of consciousness (DOC) are usually caused by a severe acquired brain injury that leads to a loss of consciousness lasting at least 24 h. Chronic DOC lasts more than three months for a non-traumatic lesion and more than 12 months with a traumatic one [3]. 

DOC etiology varies from traumatic brain injury (TBI), to intracerebral hemorrhage, and ischemic stroke or anoxia of cardiac or pulmonary insufficiency origin [5]. The level of consciousness categorizes DOC as the following: coma, if a complete absence of arousal and awareness is present [6]; vegetative state (VS), also defined as unresponsive wakefulness syndrome (UWS), characterized by arousal without awareness with a cycle of eye closure and opening similar to the sleep and waking phase in healthy subjects [7]; and minimally conscious state (MCS), when patients present with reproducible or sustained slight consciousness of self or their environment [8,9]. Despite the above definitions, the differential clinical diagnosis between UWS and MCS patients is still debated, and often leads to misdiagnosis [10,11]. 

Even though a definite timeline has been identified for the recovery of consciousness, the possibility of its reappearance is significantly reduced after 1.5 years for TBI and after 6 months for IH and IS or other causes of the disease [12]. Consequently, several attempts have been made to find methods for predicting the recovery of consciousness. 

Three prognostic factors have been positively correlated to outcomes: young age, traumatic brain injury (TBI) etiology, and low neurological impairment as MCS [13,14].

The literature has also paid attention to the analysis of sleep patterns for a better differentiation between UWS and MCS [15]. Recently the European Academy of Neurology [16] published international guidelines for a better diagnosis of coma and other disorders of consciousness through the investigation of sleep patterns, such as slow-wave and REM, and the study of the EEG using machine learning methods and artificial intelligence (AI). In addition, the management of sleep disorders in DOC patients is a increasingly hot topic and deserves a careful diagnosis, in order to allow the most accurate prognosis and best medical treatment possible [17,18]. It is noteworthy that effective treatments for sleep disorders may either indirectly or directly favor consciousness recovery [19,20].

In fact, the diagnosis of sleep patterns in DOC has received increasing attention, as it may help to evaluate the prognostic outcomes of recovery of consciousness [21]. The assessment of sleep patterns in DOC patients is still a controversial issue. Indeed, it is believed that the current criteria of the American Academy of Sleep Medicine (AASM) or Rechtschaffen and Kales (R&K) are not applicable to DOC patients, due to different electroencephalographic (EEG) and polysomnographic (PSG) patterns in patients with severe brain injury [22]. However, EEG and PSG remain the only repetitive, safe, non-invasive, and economic method to assess electric brain patterns and for sleep studies in healthy subjects and DOC patients [23]. 

An EEG is the expression of corticothalamic integrity [24], considered as the core of wakeful consciousness [1]. Usually, more severely affected DOC patients show a generalized slowing in the theta or delta range at EEG, with abnormalities such as epileptiform activity, burst-suppression, and alpha-coma pattern [25]. In MCS patients, sleep has similar neurophysiological characteristics to healthy patients, with circadian changes of EEG, electrooculogram (EOG), and muscle tone, whereas UWS patients show only EEG/behavioral signs of opening and closing their eyes. However, the two disorders are often misdiagnosed. 

Recently, De Salvo et al. [26] applied a Neurowave device to monitor the event-related potentials (ERPs) evoked by neurosensory stimulation in DOC patients. They found that the absence of an ERP component could be a distinctive marker between vs. and MCS patients [26]. On the other hand, the lack of attention period and collaboration of DOC patients limits the use of evoked potentials during the awake state, leading to false-negative results [27,28].

To date, PSG has not been able to demonstrate specificity in diagnosing and predicting consciousness [27,28]. Moreover, studies on the treatment of sleep disorders in DOC are inconclusive, due to the variable use of pharmacological and non-pharmacological interventions that did not specifically study for sleep disorders [29]. 

The aim of this review was to investigate the anatomo-physiological basis of the sleep/wake cycle, as well as the main sleep patterns and sleep disorders in patients with DOC. Some advice regarding the complex diagnosis and management of this underrated problem is also provided.

## 2. General Characteristics of Sleep Pattern

Sleep is studied worldwide by researchers and clinicians using PSG, which includes the study of EEG, electromyography, electrooculography, electrocardiography, pulse oximetry, airflow, and respiratory strength [30]. Normally, sleep is divided into two stages, as first defined by Rechtschaffen and Kales in 1968 [31], and later modified by the American Association of Sleep Medicine [32]: NREM sleep and REM sleep. These stages are further divided into cycles that range from stage 1 to stage 3 of NREM (N1–N2–N3), ending in REM sleep and wakefulness (W) with a duration of 90–110 min/cycle; and each stage lasting 5–15 min. Every stage is characterized by a specific EEG pattern [33]: delta waves and, sometimes, spindles in the N3 stage (the deepest stage of sleep); spindle and K-complex in the N2 stage; theta waves in the N1 stage; and theta waves with sawtooth waves and rapid eye movements in the REM stage; meanwhile alpha waves appear in the W stage (Figure 1). 

Recently, because of the time-consuming nature of the analysis of sleep recordings, automatic sleep stage classification has used depth network structures for sleep stage scoring, such as convolutional neural networks [34] and recurrent neural networks (RNNs) [35]. More recently, a method based on ICA-Relief-F has been proposed for the Sleep-EDF database [36,37]. However, a standard approach and procedure is still not available, to allow comparing the results of different studies. For these reasons, most of the studies in the field rely on the visual inspection of PSG recordings, which has severe limitations in terms of reproducibility and cost [18]. Normally, an EEG is characterized by more than 100 standard spindles and a full sleep–wave cycle. To better study the sleep stages, the time of sleep is conventionally divided into epochs, a continuous period of 30 s.

### 2.1. Anatomy and Physiology of the Sleep–Wake Cycle

The neuronal system that regulates the sleep–wake cycles is extended from the medulla, the brainstem, and hypothalamus up into the basal forebrain, which allows the fusing of the sleep–wake cells. The hypothalamus regulates sleep and wakefulness through the ventrolateral preoptic nucleus (VLPO), facilitated by the gamma-aminobutyric acid (GABA) and galanin neurons regulating normal sleep. In addition, the posterior lateral hypothalamus is facilitated by orexin/hypocretin neurons that regulate wakefulness, inhibiting the VLPO neurons [38].

Several neurotransmitters are released during the sleep–wake cycle, such as catecholamines, serotonin, and acetylcholine. Moreover, several factors specifically influence the wakefulness, such as olfactory, visual, vestibular, and other neuromodulators. [39]. The reticular activating system (RAS) has been shown to be the arousal system maintaining consciousness. The RAS, after receiving the impulses from the spinal cord, conveys them to the thalamus and successively to the cerebral cortex [39]. Wakefulness is promoted by the basal forebrain, lateral hypothalamus, and tuberomammillary nucleus. In particular, the neurons in the basal forebrain containing acetylcholine promote arousal involvement of the cerebral cortex during the W phase of sleep, activating alertness [40,41]. Accordingly, lesions in this area with the neurotoxin hypocretin-2-saporin cause severe loss of sleep [41]. On the other hand, the lateral hypothalamus neurons are defined as a “wake switch” for the connections to several wake-promoting areas, such as to the adrenergic, histaminergic, dopaminergic, and cholinergic nuclei [42]. According to the studies of Constantine von Economo, the anterior hypothalamus is associated with sleep, whereas the posterior hypothalamus is correlated to awareness of the histaminergic neurons [43,44] that project into the cerebral cortex, the amygdala, and the substantia nigra, and receive input from the hypocretinergic neurons of the lateral hypothalamus and GABAergic neurons in the ventrolateral preoptic area [45,46]. The brainstem sends wake-promoting neurons from the rostral reticular formation to the forebrain, regulating the sleep–wake cycle. The neurons of the rostral pons that project into the dorsal thalamic nuclei via a dorsal pathway, fire rapidly during W and REM sleep, but decrease during SWS, and are called W-on/REM-on neurons. Another pathway is the ventral pathway that projects into the magnocellular neurons in the substantia innominata, medial septum, and the diagonal band [47], actively firing during W and becoming inactive during SWS and REM sleep. As already reported, the wake phase is activated by various arousal systems that generate different neurochemicals [48], such as histamine- [43], glutamate [49], noradrenaline (NA)- [50,51,52], dopamine (DA)- [48,53], 5-hydroxytryptamine (5-HT)- [54,55], ACh- [56], and hypocretin (HCRT)-containing neurons [57,58] (Figure 2).

### 2.2. Sleep

Sleep is activated by several structures: the suprachiasmatic nucleus, the ventrolateral preoptic nucleus (VLPO), and the basal forebrain and brainstem. At the suprachiasmatic nucleus level, sleep is modulated by two mechanisms: the homeostatic and the circadian. The homeostatic mechanism is characterized by an optimal sleep duration and intensity in a short period, which depends on the individual’s history of sleep–waking. For example, a subject deprived of sleep will show an increased number of SWSs and the recovery sleep will be characterized by a longer period of sleep and delta wave activity [59]. The circadian mechanism is controlled by the suprachiasmatic nucleus (SCN) of the hypothalamus, defined as the “master clock” of sleep, because it rearranges and coordinates the circadian rhythms of the peripheral tissues. Impairment of the SCN causes a loss of W and sleep establishment [60,61,62]. The anterior hypothalamus of the VLPO nucleus is defined as the “sleep-generating” center [63,64] because of its two nuclei, the VLPO nucleus associated with SWS, and the extended VLPO nucleus, linked to REM sleep generation [65,66] and activating the inhibitory transmitters GABA and galanin that project into the arousal neurons in the hypothalamus and the brainstem [59]. Stimulus of the basal forebrain generated sleep [67,68], inhibiting the TMN nucleus [69]. Meanwhile the brainstem sends an input to the thalamic and reticular nucleus through the pedunculopontine tegmental nuclei and the latero-dorsal tegmental nuclei [70,71], whose neurons fire during REM sleep. All these important neurobiological mechanisms can be damaged by brain injury, and this damage may account for the abnormal wake–sleep cycle in DOC patients.

## 3. Characteristics and Assessment of Sleep in UWS/MCS

Sleep characteristics in DOC patients are still controversial and not well studied, due to various factors, such as the etiology of brain damage, the kind of brain dysfunction, and the related sleep abnormalities. Severe impairment of circadian rhythms has been shown [22,72,73,74], and the interruption of the regular interval of sleep and wakefulness is possibly due to injury of the brain stem [75] or cortical dysfunctions such as in the diffuse axonal damage or gray matter abnormalities due to brain hypoxia. Consequently, it was demonstrated that normal sleep was reduced at night in MCS patients with a predominance of daytime sleepiness compared to control patients. On the contrary, UWS patients showed a more severe impairment of sleep cycle with sleep phases present only during the day [76]. It could be then hypothesized that DOCs are characterized by dysregulations involving excitatory wake-promoting and inhibitory sleep-promoting neurotransmitters, which are involved in maintaining efficient connections between high-order cortical areas and sub-cortical structures [77]. A reduction of both integration and segregation was empirically described when an alteration of the diurnal sleep–wake cycle intervened. [78,79,80]. Due to the non-conventional sleep pattern of DOC patients, the use of approved terminology and scores is sometimes not applicable and suitable. Therefore, further studies are necessary to identify specific criteria for sleep and the EEG/PSG pattern in DOC patients.

Usually, the sleep–wake cycle of DOC patients is influenced by various environmental elements, such as light, bronco aspirations, hygiene, and nocturnal enteral or parenteral therapy. All these factors can cause sleep fragmentation with frequent arousals and awakenings, and this can interfere with PSG registration and sleep assessment. Moreover, other factors such as brain neurosurgery interventions, hypothermia, skin lesions or medical advice, and recovery elements can cause interference in PSG recordings. For these reasons, an independent component analysis using the Wiener and Kalman filtering wavelet analysis has been suggested to remove the myogenic artefacts of UWS patients [81]. To achieve, better results, PSG must be recorded for 48–72 h to obtain EEG data for multimodal analysis and independent component analysis algorithms, due to excessive daytime sleepiness and lack of sleep during the night. [82,83]. The sleep patterns of K-complexes, slow oscillations, and REM-sleep have been described as potential prognostic factors in this patient population in [21,84,85]. Usually, DOC patients show altered sleep phases that are clearly abnormal, especially in UWS with respect to MCS patients [86]. In particular, it has been shown that in coma patients the sleep–wake cycle and spindles are absent. Spindles and an irregular sleep–wake cycle are rarely observed in UWS; meanwhile, they are present, even if irregular, in MCS patients, as well as in the NREM2 and slow-wave sleep (SWS) stages [87].

The presence of spindle waves shows the functional integrity of the thalamus. The appearance of slow-wave and REM sleep is a good outcome for the function of the brainstem nuclei, whereas the hypothalamic functional integrity is expressed by the circadian organization of sleep patterns [18]. The sleep–wake cycle and circadian rhythm [84,87] are also normally regulated by the levels of melatonin and orexin [88,89,90]. Melatonin is secreted from the pineal gland, and it only has a circadian rhythm with high levels during the late evening [91,92]; meanwhile, orexin neurons secreted by the lateral hypothalamic area and posterior hypothalamus [93] are secreted during wakefulness and silent during non-rapid eye movement (NREM) and rapid eye movement (REM) sleep [57]. Therefore, it was shown that improving the restoration of melatonin metabolism was related to better cognitive function, evaluated using the coma recovery scale-revised (CRS-R) score in patients with brain injury [83,94,95].

At the same time, regular secretion of urinary 6-sulphatoxymelatonin and free cortisol was linked to an upgrading of cognition and awareness in comatose patients. However, Yang et al. were not able to show any involvement of the pineal gland in the retention of awareness nor a correlation between levels of orexin and recovery of consciousness [94]. In a recent study by Mertel et al. on the sleep patterns of 32 DOC patients, the authors found that MCS patients had a sleep pattern like control subjects. Specifically, in contrast with UWS, MCS and control subjects slept more during the night than during the day. Regarding REM stage, 100% of control subjects and 88% MCS patents presented REM sleep, whereas 44% of UWS did not have REM sleep. On the other hand, spindles were absent in 62% of UWS and 21% of MCS, without any REM sleep, and with spindles in 12% and 21%, respectively. Otherwise, the UWS patients did not show a night–day sleep circle and 44% and 62% did not show REM sleep and sleep spindles. On the other hand, the number of SWS was practically the same in all groups [76].

The reappearance of the eye-opening periods, of the circadian rhythm, and of the behavioral sleep–wake cycle represent signals of a return of awareness and arousal from a coma state to MCS [96]. However, if significant damage of the cerebral cortex exists, such as a loss of the cortical neurons or a reduction in the number of interneuron connections in the cerebral cortex, axonal damage with degeneration of cortical neurons, or a reduction in the activity of the caudal group of the nuclei of the basal forebrain, the PSG is insufficient. Loss of cortical neurons and their connections is converted into a slowing down in cortical rhythm and the absence of high-frequency activity of subcortical structures and synchronized postsynaptic excitatory potentials of cortical neurons [97]. Thus, sleep spindles are absent in EEG recordings. Therefore, the presence of spindles designates a preserved cerebral cortex and a favorable prognosis [98]. Spindles represent the preservation of thalamocortical connectivity. Therefore, they have been considered as a predicted sign for plasticity and of early behavioral awakening in comatose patients [99] and when absent, a poor outcome should be considered [100].

Regarding the REM phase, even though it has been shown that the REM phase in UWS patients was significantly reduced, there was no correlation between the frequency of these activities and recovery of the clinical condition. [101]. Moreover, an impaired REM phase is a signal of compromised brain stem mechanisms. In one-third of UWS patients and in more than half of MCS patients, preserved sleep spindles, rapid eye movement, and slow-wave sleep with favorable outcomes were shown, but only when a high quality and quantity of sleep standard spindles were present [18,84]. To allow the best neurological prognosis, the EEG must be scored based on American Clinical Neurophysiology Society (ACNS) terminology [102]. The ACNS takes into consideration several EEG characteristics, such as time domain or frequency domain, that provided effective and comprehensible and applicable indices to describe brain electrical activities under different states of consciousness. Therefore, a standard EEG recording classified according to the ACNS terminology may improve the capability for neurological prognosis [22,102]. Recently, Wielek et al. investigated a multivariate machine learning technique to categorize sleep/wake stages in DOC patients, which may represent an alternative to the standard PSG [103].

## 4. Treatment

Despite the importance of sleep to human beings, the treatment of sleep disorders in patients with DOC is still overlooked in both research and clinical practice. What is more, it has been shown that therapeutic schemes to improve sleep and sleep–wake cycles may improve recovery in DOC [16,96]. Therefore, this issue deserves attention in the near future.

A recent review [29] on pharmacological and non-pharmacological treatments focused on two kinds of sleep disorder: sleep-related breathing disorders [100,101], and circadian rhythm sleep–wake disorders [73,104,105,106,107,108], and both can affect DOC patients. One of the major pharmacological treatments that has proven effective in sleep disorders is modafinil. As a dopamine reuptake inhibitor, it also disregards the expression of orexin, improving the circadian rhythm regulation and reducing excessive daily sleepiness [109,110]. Moreover, an increased activation of the frontal parietal control (FPC) and the dorsal attention network (DAN) with modulation of the functional connectivity of the resting state of networks, including the FPC, the DAN, and the extrastriate visual system [67] has been described. The effect of promoting network integration was associated with improvements in cognitive performance, both in clinical populations [111,112] and healthy individuals [113,114]. Therefore, modafinil could be used in DOC patients, either to treat sleep disorders, daytime drowsiness associated with OSA [115], narcolepsy [116], and shift work sleep disorder [117], as well as to potentially promote recovery of consciousness [118,119,120,121].

Other psychostimulant drugs have been used for their capacity to recover consciousness, such as amantadine and Zolpidem [20,122,123,124], without proven effects on sleep disorders.

Locatelli et al. [104] described the appearance of sleep apnoea after the increment of Intrathecal Baclofen (ITB) daily dosage (from 450 μg/d to 600 μg/d) for spasticity in a pediatric MCS patient, which was subsequently solved with Baclofen tapering (100 μg/day). Moreover, ITB has been related to a increase in motor and cognitive functioning [105] and has a potential role in stimulating the recovery of consciousness [125,126].

### Non-Pharmacological Treatment

Non-pharmacological interventions include positive airway pressure (PAP) for breathing disorders [105], and bright light stimulation (BLS) [73] and central thalamic-deep brain stimulation (CT-DBS) [73,108] being used as treatments for circadian sleep–wake disorders. PAP has been shown to reduce sleep-apnoea phenomena, with a consequent cognitive recovery [127]. However, further studies are necessary to show the direct or indirect effectiveness of PAP on consciousness recovery in patients with DOC. BLS is based on the impact that environmental light has on the suprachiasmatic nuclei of the hypothalamus, which regulates circadian processes [128,129,130,131]. This can improve the sleep–wake cycle, even in neurodegenerative disorders [132], and the processes that support sustained attention and awareness [132,133,134]. Even though no research has shown DOC recovery through resetting circadian rhythms, light of day exposure improves at least some behaviors [135,136]. Moreover, in a recent study, Yelden et al. treated 10 DOC patients with melatonin, light, and caffeine for 5 weeks, obtaining an improvement in arousal and awareness cycle, as assessed by behavioral (CRS-R) and neurophysiological measures [129].

CT-DBS was used to elicit the circuit of the thalamus and modulate the fronto-cortical circuit, influencing the restoration of the sleep circuit (47,49,81). In particular, in MCS patients, [105,106,107] CT-DBS treatment improved the sleep pattern, with an increase in the frequency of sleep spindles during NREM-2 and during slow wave sleep (SWS) stages. Discontinuation caused a lower CRS-r score and a disruption in sleep architecture after one year (Table 1) [107].

Even though improvements have been shown in behavioral and cognitive performance, and/or an increased awareness in DOC patients, with a fully recovered consciousness in a total of 14 patients [73,108], their effects have not yet been explored in randomized controlled trials.

Therefore, we can conclude that treating sleep disorders could favor recovery of consciousness, based on the neuroanatomical and functional correlation [76,94,137,138,139,140].

## 5. Discussion

Sleep is a fundamental function of all human beings, although its importance seems underestimated for patients with DOC. With this review, we wanted to demonstrate how many different brain structures are involved in the sleep–wake cycle and, above all, to what extent these complex neural pathways can be altered by severe brain injury, leading to many sleep pattern abnormalities in DOC.

Until 2010, only a few studies had been carried out on this issue, showing the characteristics of REM sleep stage in UWS [141,142,143,144,145], such as a dysfunction in the organization of sleep and the presence of alterations of phase II of the slow-wave sleep recorded without sleep spindles in more than half of UWS patients. Between 2010 and 2013, research on sleep disorders increased, even though a clear PSG pattern was still debated [146,147,148,149].

However, as previously reported, the modality of PSG recording is fundamental for obtaining the best and most concrete results. Environmental factors such as light at night, drug therapy, and sound of the hospital, as well as movement to avoid decubitus ulcers, alter sleep during the night and confuse the interpretation of sleep abnormalities in PSG.

To better evaluate the sleep–wake cycle, a long 24–48 h PSG recording is necessary, to overcome possible environmental influences on the sleep phase, while the use of standardized methods of sleep classification permits obtaining a better accuracy of sleep study [21]. However, new criteria for PSG sleep analysis for DOC patients are still required.

A better organized sleep pattern is a good predictive value for the prognosis of functional recovery [140,150]. A well-structured EEG-sleep pattern is fundamental, because the impairment of sleep pattern is related to excessive day sleepiness and impairment of alertness, with a consequent cognitive impairment, and lack of memory and attention [38]. In fact, DOC patients often present a theta/delta pattern or epileptiform activity, while other sleep elements were reported to correlate with outcome in patients with coma and DOC [82,85,151,152]. For example, the absence of spindles, as well as REM and K-complexes of NREM sleep have been used as negative predictors of early awakening in DOC patients, especially in UWS patients [76,82,83,94,95,96,146].

There are several factors that can improve the prognosis in patients with DOC, such as etiology, age, and the time interval after brain injury, as well as GCS sub-scores, EEG signals, and various sensory evoked potentials of the patient. Regarding sleep, subjects with DOC showed that the functional integrity of the thalamus can be reflected by the recording of spindle waves. The effective functioning of the brainstem nuclei may likewise be recorded by the SWS and rapid eye movement phases [18].

In a recent systematic review of the brain measurements in DOC patients [59], the authors reported that the amount and characteristics of sleep spindle waves, REM sleep, and circadian sleep–wake cycle may be helpful in distinctive patients with MCS and UWS. The authors also emphasized the importance of a correct PSG recording and suggested the necessity of updating sleep scoring criteria and of an individualized early prognosis [18,96,152].

In addition, treatment remains a controversial issue, although the use of some drugs, including modafinil, may improve wake–sleep cycle and consciousness recovery [15,19,87,153,154,155,156].

## 6. Conclusions

Our review sought to demonstrate the importance of sleep and its abnormalities in the prediction of recovery of consciousness. In particular, spindles, SWS, and REM sleep, and circadian rhythms have been shown to be abnormal in MCS patients and UWS, raising the possibility that these elements could be used as markers in the differential diagnosis of DOC [18]. Nonetheless, an optimal recording of PSG is necessary to overcome the various hospital environmental factors that interrupt sleep in these frail individuals. Further studies are necessary, to better investigate sleep stages in DOC patients, in order to more properly manage sleep disorders and boost recovery of consciousness.

## Figures and Tables

**Figure 1 brainsci-13-00275-f001:**
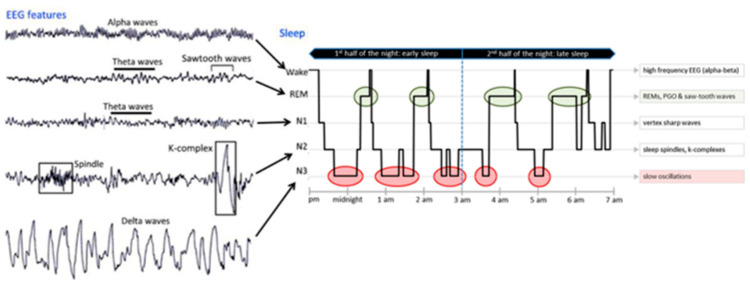
EEG pattern (on the **left**) and an 8 h recorded hypnogram (on the **right**) of normal sleep stages.

**Figure 2 brainsci-13-00275-f002:**
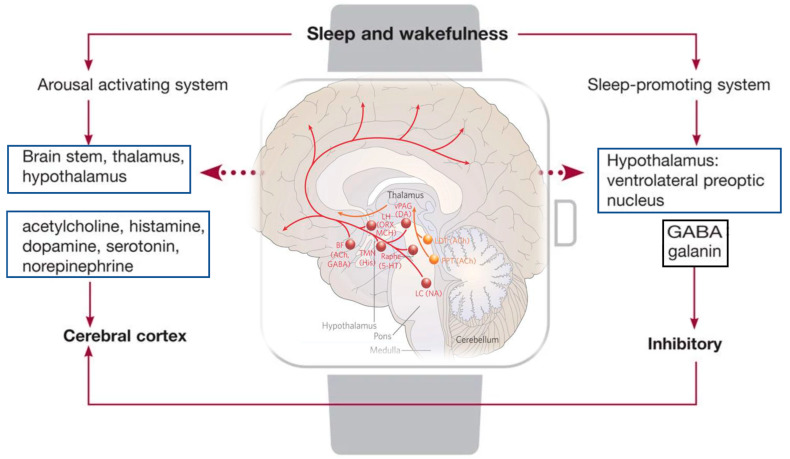
Structures and neurotransmitters responsible for arousal and sleep. In the figure: regulation of the ascending arousal system by the hypothalamic. Red lines represent the pathway of activation of the cortex, to enable the handling of inputs from the thalamus. The monoaminergic structures involved in this process are the locus coeruleus (LC), by noradrenaline (NA); the dorsal and median raphe nuclei, by serotonin (5-HT); the A10 cell group, by dopamine (DA); and the tuberomammillary nucleus (TMN), by histamine (His). Moreover, peptidergic structures are involved, such as the lateral hypothalamus (LHA), by orexin (ORX), or melanin-concentrating hormone (MCH) and basal forebrain (BF) neurons that contain γ-aminobutyric acid (GABA) or acetylcholine (Ach). The orange lane represents the input to the thalamus from cholinergic (ACh) structures, such as the upper pons, the pedunculopontine (PPT), and laterodorsal tegmental nuclei (LDT).

**Table 1 brainsci-13-00275-t001:** Main pharmacological and non-pharmacological treatments for DOC sleep-related disorders.

Treatment	Sleep Disorder	Effect
Modafanil	excessive sleepiness and circadian sleep–wake disorders	significant increase and stabilization of sleep–wake periods.
BLS	circadian sleep–wake disorders	circadian restoration of sleep–wake periods
CT-DBS	circadian sleep–wake disorders	modification in the cortical activity restoration of good sleep architecture and of most physiological NREM-2 and SWS stages of sleep,
PAP	sleep-related breathing disorders	reduction of sleep-apnoea
ITB	sleep-related breathing disorders	tapering ITB daily dose improved central sleep apnoea
Melatonin, light and caffeine	circadian sleep–wake disorders	circadian rebuilding of sleep–wake cycle

Positive airway pressure (PAP) for breathing disorder, bright light stimulation (BLS) and central thalamic-deep brain stimulation (CT-DBS), Intrathecal Baclofen (ITB).

## Data Availability

Not applicable.

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
