# Peer review of "Sleep in Disorders of Consciousness: A Brief Overview on a Still under Investigated Issue"

_brainsci, 2023, doi:10.3390/brainsci13020275_

Round 1
Reviewer 1 Report
This review by Raciti et al is a very comprehensively detailed and particularly educational review for any expert or new beginner within the sleep field. This reviewer does not have any corrections and proposes to accept the manuscript.
Author Response
Thank you for this positive evaluation of the paper
Reviewer 2 Report
It is my pleasure to review this article. The article discussed an interesting topic, Sleep, with emerging importance in the field of DoC. The authors well presented the background of the issue in the introduction. Then, they introduced the basics of sleep/wake cycle, discussed electrophysiological characteristics of sleep in DoC, and potential treatment options for sleep disorders in DoC based on the current understanding. I have the following comments and suggestions.
Would recommend including the following reference for the statement in Line 50-52 “Despite the above definitions, the differential clinical diagnosis between UWS and MCS patients is still debated, and usually leads to misdiagnosis [10].”
[] Zhang B, Huang K, Karri J, O'Brien K, DiTommaso C, Li S. Many Faces of the Hidden Souls: Medical and Neurological Complications and Comorbidities in Disorders of Consciousness. Brain Sci. 2021;11(5):608. Published 2021 May 10. doi:10.3390/brainsci11050608
Would be great if the authors could use a diagram to demonstrate the neuroanatomy and physiological processes mentioned in 2.1 and 2.2, but up to the authors’ discretion.
Please rephrase this sentence in Line 190-193. It’s not understandable. “Consequently, MCS patients show reduced sleep at night and daytime sleepiness as compared to control patients, meanwhile a more severe impairment without a sleep and wakefulness phase and, usually with a sleep present only during the day, caracterarize UWS patients [76].”
May consider incorporating the content in Line 223-228 in the section of 2.1. Anatomy and Physiology of Sleep/Wake Cycle.
Please explain more about the sentence in Line 265-267. “To better identify the best neurological prognosis, the EEG has to be scored based on the American Clinical Neurophysiology Society (ACNS) terminology [104].” Because previously, the authors stated that “it is believed that the current criteria of the American Academy of Sleep Medicine (AASM) or Rechtschaffen and Kales (R&K) are not applicable to DOC patients due to different electroencephalographic (EEG) and polysomnographic (PSG) patterns in patients with severe brain injury [21]” in Line 70-74.
Please correct the information in Line 292 and 293 regarding “(ITB) daily dosage (from 450 _g/die to 600 _g/d)” and “Baclofen tapering (100 _g/day).”
For treatment, there are newer studies on using melatonin and natural light that are worth mentioning.
[] Yelden K, James LM, Duport S, et al. A simple intervention for disorders of consciousness- is there a light at the end of the tunnel?. Front Neurol. 2022;13:824880. Published 2022 Jul 22. doi:10.3389/fneur.2022.824880
[] Wang J, Di H. Natural light exposure and circadian rhythm: a potential therapeutic approach for disorders of consciousness. Sleep. 2022;45(7):zsac094. doi:10.1093/sleep/zsac094
Many of the references are not up to date. Would suggest the authors to update the references in order to provide the readers with the latest information. Additionally, the listed Ref 159 was not cited in the article and the format is incomplete. Please thoroughly examine all references.
Overall, the logic and layout of the article are good. The content is understandable and carries clinically meaningful information. I would suggest the authors consider using shorter sentences. Long sentences would not be ideal for readability.
Thanks!
Author Response
Response to the reviewer
- Would recommend including the following reference for the statement in Line 50-52 “Despite the above definitions, the differential clinical diagnosis between UWS and MCS patients is still debated, and usually leads to misdiagnosis [10].”
[] Zhang B, Huang K, Karri J, O'Brien K, DiTommaso C, Li S. Many Faces of the Hidden Souls: Medical and Neurological Complications and Comorbidities in Disorders of Consciousness. Brain Sci. 2021;11(5):608. Published 2021 May 10. doi:10.3390/brainsci11050608
Thank you for your recommendation. We added the suggested reference in the text.
- Would be great if the authors could use a diagram to demonstrate the neuroanatomy and physiological processes mentioned in 2.1 and 2.2, but up to the authors’ discretion.
Thank you very much for your suggestion. We added the requested figure in the text.
- Please rephrase this sentence in Line 190-193. It’s not understandable. “Consequently, MCS patients show reduced sleep at night and daytime sleepiness as compared to control patients, meanwhile a more severe impairment without a sleep and wakefulness phase and, usually with a sleep present only during the day, caracterarize UWS patients [76].”
Thank you for your suggestion. We rephrased the sentence.
- May consider incorporating the content in Line 223-228 in the section of 2.1. Anatomy and Physiology of Sleep/Wake Cycle.
We kindly appreciate your interesting observation. However, the lines 223-228 tried to explain the complexity of physiology of Sleep in coma/UWS and MCS patients linking to the previous brief paragraph of sleep characteristics of disorders of consciousness (DOC).
- Please explain more about the sentence in Line 265-267. “To better identify the best neurological prognosis, the EEG has to be scored based on the American Clinical Neurophysiology Society (ACNS) terminology [104].” Because previously, the authors stated that “it is believed that the current criteria of the American Academy of Sleep Medicine (AASM) or Rechtschaffen and Kales (R&K) are not applicable to DOC patients due to different electroencephalographic (EEG) and polysomnographic (PSG) patterns in patients with severe brain injury [21]” in Line 70-74.
Thank you very much for your observation.
It has been shown that a standard EEG recording in patients with severe DOC in the early period of the recovery to the Intensive Rehabilitation Unit (IRU) classified according to an EEG score based on the ACNS terminology improved neurological prognosis. In particular, the ACNS taken into considerations several EEG characteristics as time domain or frequency domain that provided effective and comprehensible and applicable indices to describe brain electrical activities under different conscious states (Scarpino M, Lolli F, Hakiki B, Lanzo G, Sterpu R, Atzori T, Portaccio E, Draghi F, Amantini A, Grippo A;, «Intensive Rehabilitation Unit Study Group of the IRCCS Don Gnocchi Foundation, Italy. EEG and Coma Recovery Scale-Revised prediction of neurological outcome in Disorder of Consciousness patients patients.,» Acta Neurol Scand. 2020 Sep;142(3):221-228. doi: 10.1111/ane.13247. Epub 2020 Apr 14. PMID: 32219851; Wislowska, M.; del Giudice, R.; Lechinger, J.; Wielek, T.; Heib, D.P.J.; Pitiot, A.; Pichler, G.; Michitsch, G.; Donis, J.; Schabus, M., «Night and day variations of sleep in patients with disorders of consciousness.,» Sci. Rep. 2017, 7, 266).
On the other hand, the classification according to the AASM criteria may often be confusing and inconsistently on staging. From the pathological eye movements to spasms, and uncommon topographies or presence of trademarks of different sleep stages caused a sleep-classification improbable. Polysomnography in DOC is still under investigation and has to be cautiously applied as a diagnostic test because of the imperfectly PSG information in cases of significant damage to the cerebral cortex (as reported in 249-255 sentences) (Nekrasova J, Kanarskii M, Yankevich D, Shpichko A, Borisov I, Pradhan P, Miroshnichenko M. , «Retrospective analysis of sleep patterns in patients with chronic disorders of consciousness.,» Sleep Med X. 2020 Aug 28;2:100024. doi:10.1016/j.sleepx.2020.100024; Wislowska, M.; del Giudice, R.; Lechinger, J.; Wielek, T.; Heib, D.P.J.; Pitiot, A.; Pichler, G.; Michitsch, G.; Donis, J.; Schabus, M., «Night and day variations of sleep in patients with disorders of consciousness.,» Sci. Rep. 2017, 7, 266.)
To better clarify this point we added the following sentences in the text (lines 269-273):
“The ACNS takes into considerations several EEG characteristics as time domain or frequency domain that provided effective and comprehensible and applicable indices to describe brain electrical activities under different conscious states. Therefore, a standard EEG recording classified according to the ACNS terminology improved neurological prognosis [21]; [104].”
Recently, Wielek et al investigate a multivariate machine learning technique to categorise
sleep/wake stages in DOC patients that may represent an alternative to the standard PSG (Wielek T, Lechinger J, Wislowska M, et al. Sleep in patients with disorders of consciousness characterized by means of machine learning. PLoS One. 2018;13(1):e0190458. Published 2018 Jan 2. doi:10.1371/journal.pone.0190458)
- Please correct the information in Line 292 and 293 regarding “(ITB) daily dosage (from 450 _g/die to 600 _g/d)” and “Baclofen tapering (100 _g/day).”
Thank you for your observation. We correct as “(ITB) daily dosage (from 450 μg/die to 600 μg/d)” and “Baclofen tapering (100 μg/day).
- For treatment, there are newer studies on using melatonin and natural light that are worth mentioning.
[] Yelden K, James LM, Duport S, et al. A simple intervention for disorders of consciousness- is there a light at the end of the tunnel?. Front Neurol. 2022;13:824880. Published 2022 Jul 22. doi:10.3389/fneur.2022.824880
[] Wang J, Di H. Natural light exposure and circadian rhythm: a potential therapeutic approach for disorders of consciousness. Sleep. 2022;45(7):zsac094. doi:10.1093/sleep/zsac094
Thank you for your suggestion. To the already cited bright light stimulation, we added information about the suggested works, even in table 1:
“Even though no research has shown DOC recovery by resetting circadian rhythms mimicking, a light of day exposure improves only behaviors [Angerer M, et al. From dawn to dusk—mimicking natural daylight exposure improves circadian rhythm entrainment in patients with severe brain injury. Sleep. 2022;45(7). doi:10.1093/sleep/zsac065]. Moreover, in a recent study of Yelden et al treated 10 DOC patients with melatonin, light and caffeine for 5 weeks obtaining an improvement on arousal and awareness cycle assessed by behavioral (CRS-R) and neurophysiological measures. Yelden K, James LM, Duport S, et al. A simple intervention for disorders of consciousness- is there a light at the end of the tunnel?. Front Neurol. 2022;13:824880. Published 2022 Jul 22. doi:10.3389/fneur.2022.824880”
- Many of the references are not up to date. Would suggest the authors to update the references in order to provide the readers with the latest information. Additionally, the listed Ref 159 was not cited in the article and the format is incomplete. Please thoroughly examine all references.
Thank you for your observation. We added some references to try to improve the overall information.
Overall, the logic and layout of the article are good. The content is understandable and carries clinically meaningful information. I would suggest the authors consider using shorter sentences. Long sentences would not be ideal for readability.
We reviewed the paper accordingly.
Round 2
Reviewer 2 Report
The authors have adequately addressed my questions and comments. One minor correction may be still needed. Line 319: "from 450 _μg/die."